# Absolute Quantification of Hepatitis B Core Antigen (HBcAg) Virus-like Particles and Bound Nucleic Acids

**DOI:** 10.3390/v16010013

**Published:** 2023-12-21

**Authors:** Angela Valentic, Nicola Böhner, Jürgen Hubbuch

**Affiliations:** Institute of Process Engineering in Life Sciences—Section IV: Biomolecular Separation Engineering, Karlsruhe Institute of Technology (KIT), 76131 Karlsruhe, Germany; angela.valentic@kit.edu (A.V.); nicola.boehner@kit.edu (N.B.)

**Keywords:** virus-like particles, nucleic acid extraction, silica spin column, RP-HPLC, HBcAg, analytical characterization, gene delivery

## Abstract

Effective process development towards intensified processing for gene delivery applications using Hepatitis B core Antigen (HBcAg) virus-like particles (VLPs) relies on analytical methods for the absolute quantification of HBcAg VLP proteins and bound nucleic acids. We investigated a silica spin column (SC)-based extraction procedure, including proteinase K lysis and silica chromatography, for the absolute quantification of different species of nucleic acids bound to HBcAg VLPs analyzed by dye-based fluorescence assays. This revealed load-dependent nucleic acid recoveries of the silica-SC-based extraction. We also developed a reversed-phase high-performance liquid chromatography (RP-HPLC) method to separate and quantify the HBcAg proteins and the bound nucleic acids simultaneously without prior sample treatment by dissociation reagents. The method demonstrated sufficient linearity, accuracy, and precision coefficients and is suited for determining absolute protein and nucleic acid concentrations and HBcAg protein purities at various purification stages. Both the silica-SC-based extraction and the RP-based extraction presented overcome the limitations of analytical techniques, which are restricted to relative or qualitative analyses for HBcAg VLPs with bound nucleic acids. In combination with existing analytics, the methods for an absolute quantification of HBcAg VLPs and bound nucleic acids presented here are required to evaluate downstream purification steps, such as the removal of host cell-derived nucleic acids, concurrent protein loss, and efficient loading with therapeutic nucleic acids. Hence, the methods are key for effective process development when using HBcAg VLP as potential gene delivery vehicles.

## 1. Introduction

In recent years, virus-like particles (VLPs) have gained significant attention due to their importance as potential vaccines and gene delivery vehicles [1,2,3,4,5,6,7]. Among these VLPs, Hepatitis B core antigen (HBcAg) has emerged as a promising candidate for various biomedical applications [8,9,10,11,12,13]. For gene delivery purposes, the naturally occurring nucleic acid binding region of the HBcAg but also biologically engineered binding regions are often utilized to effectively encapsulate therapeutic nucleic acids (NA_ther_) [8,10]. During intracellular formation of HBcAg VLPs in *E. coli* [14], host cell-derived nucleic acids (NA_hc_) are bound to the nucleic acid binding region and encapsulated in the VLPs [8,15,16,17]. During the subsequent downstream purification process and before reloading the VLPs with NA_ther_, it is crucial to remove entrapped and bound NA_hc_. For this purpose, techniques such as enzymatic treatment [18], affinity chromatography [15], alkaline treatment [16], and lithium chloride precipitation [8] are investigated in the literature. After successful depletion of the host cell nucleic acids, the VLPs are often loaded with the respective NA_ther_ by direct mixing [10,16]. To evaluate the successful depletion of NA_hc_ and possible HBcAg protein loss during removal as well as subsequent effective loading of the VLPs with NA_ther_ on the basis of, e.g., the encapsulation efficiency or payload of the VLPs, a precise absolute quantification of HBcAg VLPs and VLP-bound nucleic acids is needed.

However, absolute quantification of HBcAg VLPs and the bound NA_hc_ remains challenging. UV/Vis spectroscopy is a common tool to quantify proteins and nucleic acids, utilizing the absorption maxima of proteins at 280 nm and nucleic acids at 260 nm. In combination with high-performance liquid chromatography (HPLC) methods, UV spectra allow for the quantification of different HBcAg VLP protein species (capsids and dimers) and impurities such as host cell proteins (HCPs) and free nucleic acids by size exclusion chromatography [19,20,21,22]. Additionally, reversed-phase (RP) chromatography is commonly used to quantify different VLPs. In this case, the sample is subjected to a preliminary treatment with dissociation buffer containing guanidine-HCl and DTT [23,24] or zwitterionic detergent [25]. However, a specific RP-HPLC method for the quantification of HBcAg VLPs with bound nucleic acids has not been published in the literature.

UV/Vis spectroscopy is inherently unable to absolutely quantify HBcAg VLP proteins with bound nucleic acids without prior quantitative separation of proteins and nucleic acids, e.g., by chromatography. In the absence of a successful separation, the overlapping signals of the proteins and nucleic acids in the relevant wavelength region of the UV spectra only allow for a relative quantification. Relative analysis of the A260/A280 ratio of HBcAg proteins and bound NA_hc_ has been used to correlate HBcAg VLP constructs with different lengths of the nucleic acid binding region to particle loading with NA_hc_ [26]. It can also be used to evaluate the purity of the VLPs after the depletion of NA_hc_ [8,15,27]. For the loading of HBcAg VLPs with NA_ther_ of known nucleotide sequences, a method to approximate the absolute amounts of protein and nucleic acid was proposed in the literature [28] and used to estimate protein and nucleic acid composition [10,16]. However, this method is not suited for the precise quantification of VLP proteins and (i) non-specific NA_hc_. Nor can it be applied to (ii) mixtures of residual undesired NA_hc_ and NA_ther_ or (iii) mixtures of various types of NA_ther_. This information is, however, required for evaluating the successful depletion of NA_hc_ and possible HBcAg protein loss during the removal, as well as the subsequent effective loading of the VLPs with NA_ther_.

While dye-based fluorescence assays are commonly used for nucleic acid quantification, they can be sensitive to contaminants that interfere with the binding of the dye to its target [29]. In the literature, a RiboGreen assay was used for the quantification of RNA in murine leukemia VLPs [30]. Before quantification, the RNA was extracted according to a nucleic acid extraction protocol including proteinase K lysis and phenol-chloroform extraction. In contrast, no prior extraction of DNA was required in a study by Effio and Hubbuch [14], as it was not bound to the VLPs and did not interfere with the binding of the dye to the DNA. Here, a PicoGreen assay was used for the quantification of DNA impurities in HBcAg VLP purification. However, for HBcAg VLPs with a nucleic acid binding region for gene delivery and thus bound nucleic acids, appropriate nucleic acid extraction seems to be necessary to quantify the nucleic acids precisely. This can be concluded from the conflicting results of the relative analysis of the A260/A280 ratio with UV spectroscopy and a dye-based fluorescence assay without prior extraction of the HBcAg VLPs and bound NA_hc_ [27].

A variety of methods, such as phenol-chloroform extraction, anion exchangers, silica membranes, cesium chloride density gradient centrifugation, and polyethylene-glycol (PEG) precipitation, are used for the extraction of nucleic acids in molecular biology [31,32,33,34,35]. Commonly used methods to specifically separate protein contaminants from nucleic acids are phenol-chloroform extraction, PEG precipitation, or silica column-based purification, coupled with a preceding proteinase K digestion, if proteins are bound to the nucleic acids. However, phenol-chloroform extraction is rarely used today because of the caustic and harmful chemicals involved [31,36]. PEG precipitation is characterized by a complex interplay of PEG lengths and the type of nucleic acids precipitated [31,35], and is also known to precipitate proteins [37]. While silica column-based purification provides reproducible results, it appears possible to overcome nucleic acid recovery issues through a yield increase through recirculation of the extracts over anion column membranes [31]. For their use in molecular biology, these methods do not necessarily require high purity and recovery. The potential loss of nucleic acid during the extraction process is often irrelevant for the further processing of the nucleic acids or is compensated by reproducing the extracted nucleic acids by polymerase chain reaction prior to analysis [31].

These methods might be used as nucleic acid extraction methods for the absolute quantification of nucleic acids in HBcAg VLP process development. However, for accurate quantification of nucleic acids bound to VLP proteins, the nucleic acid solution must be of high purity after the extraction, and the loss of nucleic acids during these procedures must be minimized. Indeed, proteinase K treatment followed by phenol-chloroform extraction is used to isolate RNA from HBcAg VLPs in preparation for native agarose gel electrophoresis (NAGE) [10] and to extract RNA from murine leukemia VLPs prior to the RiboGreen assay [30], but purity and recovery of the extraction procedures are not verified prior to analysis, and the analysis thus has to be considered semi-quantitative. Beyond that, for HBcAg VLPs with bound nucleic acids, a nucleic acid extraction procedure expressing sufficient purity and specified nucleic acid recoveries for accurate quantification of NA_hc_ or NA_ther_ by dye-based fluorescence assays to evaluate the successful depletion of NA_hc_ and possible HBcAg protein loss during the removal, as well as the subsequent effective loading of the VLPs with NA_ther_, still remains to be developed.

In this work, we present a comprehensive investigation into a silica spin column (SC)-based extraction for the absolute quantification of NA_hc_ and NA_ther_ bound to HBcAg VLPs by dye-based fluorescence assays and newly developed an RP-HPLC method for the separation and simultaneous quantification of HBcAg VLP proteins and bound nucleic acids (Figure 1). For the silica-SC-based extraction, lysis and silica-SC chromatography were investigated to achieve optimal purity and recovery of the nucleic acids to be quantified. The need for nucleic acid extraction prior to quantification by dye-based fluorescence assays was demonstrated by comparing samples before and after the silica-SC-based extraction. The suitability of the silica-SC-based extraction for further processing using the RiboGreen assay or quantitative PCR (qPCR) was verified (Table 1). Moreover, we developed a novel RP-based extraction method for simultaneous quantification of HBcAg VLP proteins and NA_hc_ (Figure 1). The method separates HBcAg VLP proteins and bound nucleic acids due to the lysis of the VLPs by the RP-HPLC loading mobile phase, with the latter being quantified in the flow-through and the former in the gradient elution of the RP-HPLC. The method was applied to quantify the absolute amounts of NA_hc_ in HBcAg VLP constructs with different lengths of nucleic acid binding regions, correlating to the relative results described earlier [26]. The proposed methods turned out to be suited for the absolute quantification of HBcAg VLP proteins and bound nucleic acids. This is required to evaluate the depletion of NA_hc_ and possible HBcAg protein loss during the process, as well as the effective loading of the VLPs with NA_ther_ in terms of the encapsulation efficiency or payload of the HBcAg VLPs for their use as gene delivery vehicles.

**Table 1 viruses-16-00013-t001:** Overview of approaches and utilized analytics for the investigated silica-SC-based extraction, including Proteinase K lysis and silica spin column chromatography, characterization, and analysis. B/E: bind and elute; NAGE: native gel electrophoresis.

Silica-SC-Based Extraction	Approach	Analytics	Data
**Process**	protein lysis	SDS-PAGE	Figure 2
	B/E: nucleic acid adsorption	NAGE	Figure 3
	B/E: nucleic acid elution	NAGE, RiboGreen	Figure 4
**Characterization**	recovery	RiboGreen, qPCR	Figure 5
	with/without lysis and extraction	RiboGreen	Table 2
**Analysis**	lambda DNA/16S RNA	RiboGreen	Figure 6
	enzymatic treatment	RiboGreen, PicoGreen	Table 3

**Figure 1 viruses-16-00013-f001:**
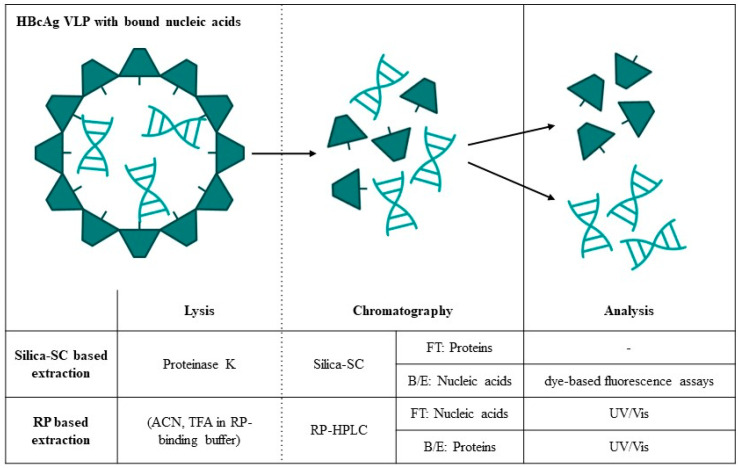
Lysis, chromatography, and analysis techniques investigated in this work for the absolute quantification of HBcAg VLP proteins and bound nucleic acids. B/E: bind and elute; FT: flow-through; HBcAg: Hepatitis B core Antigen; HPLC: high-performance liquid chromatography; RP: reversed phase; SC: spin column; VLP: virus-like particle.

## 2. Materials and Methods

### 2.1. Buffers, VLPs, and DNA

If not stated otherwise, all chemicals were purchased from Merck (Darmstadt, Germany). Solutions and buffers were prepared with ultrapure water (PURELAB Ultra, Veolia Water Solutions & Technologies, Paris, France and aqueous buffers were filtered through a 0.2 µm pore-size cellulose acetate filter (Pall Corporation, Port Washington, NY, USA). Buffers were pH-adjusted with 32% HCl or 4 M NaOH. HPLC-grade acetonitrile (ACN) was purchased from Avantor (Radnor, PA, USA) and trifluoroacetic acid (TFA) from Thermo Fisher Scientific (Waltham, MA, USA).

HBcAg VLPs were produced and purified as described earlier [26]. Constructs Cp149, Cp154, Cp157, Cp164, Cp167, and Cp183 with different lengths of nucleic acid binding regions and different amounts of bound host cell-derived nucleic acids (NA_hc_) [26] after re-dissolution or CaptoCore 400 (CC) purification were used for experiments. Prior to all experiments, HBcAg VLPs were present in purification buffer consisting of 50 mM Tris and 150 mM NaCl at pH 7.2. The dsDNA used, with a length of 720 bp, was produced by amplification of a pKR-based vector including an eGFP-encoding gene sequence, using 5′-ATGGTGAGCAAGGGCGAG-3′ as a forward primer and 5′-TTACTTGTACAGCTCGTCCA-3′ as a reverse primer. Amplification was performed with PCRBio HiFi polymerase (Nippon Genetics Europe GmbH, Düren, Germany). The PCR product was purified by native gel electrophoresis and extracted using a Wizard SV gel and a PCR clean-up kit (Promega, Madison, WI, USA).

### 2.2. Silica Spin Column-Based Extraction

The silica spin column (SC)-based extraction consists of (i) enzymatic lysis at 56 °C for 15 min or 60 min using 12 U proteinase K, (ii) addition of ethanol and adsorption of the nucleic acids on a silica SC, centrifugation and disposal of the flow-through, (iii) washing with a wash buffer, and (iv) elution of the nucleic acids from the silica-SC by addition of a total of 100–200 µL of nuclease-free water (New England Biolabs, Ipswich, MA, USA), followed by centrifugation. In the following, the terminology ‘silica-SC-based extraction’ comprises proteinase K lysis and silica-SC chromatography. The materials used for the silica-SC-based extraction were included in the EasyPure Viral DNA/RNA Kit (TransGen Biotech, Beijing, China). To achieve optimal purities and recoveries of the extracts, (i) lysis, (ii) adsorption, and (iv) elution procedures were investigated. Lysis times of 15 min and 60 min, binding procedures without and with 5-fold and 20-fold recirculation of the flow-through, and elution with 4-fold and 8-fold 25 µL and 1-fold 200 µL elution volumes were evaluated. To analyze different elution procedures and the recovery of nucleic acids for different loadings, NA_hc_-stocks with a known concentration were produced. For this, CC-purified Cp157 was extracted, and the nucleic acid concentration was determined by a RiboGreen assay. Silica-SC-based extraction results were obtained with the materials included in the EasyPure Viral DNA/RNA Kit. Prior to silica-SC-based extraction, the nucleic acid content of the VLP samples was estimated by UV/Vis absorbance measurements using a NanoDrop^TM^ 2000c UV/Vis spectrophotometer (Thermo Fisher Scientific, Waltham, MA, USA). Samples were diluted to a maximum of 25 ng/µL nucleic acids, if not stated otherwise.

### 2.3. RP-Based Nucleic Acid and Protein Extraction

Analytical reversed-phase chromatography was performed with a TSKgel Protein C4-300 column (3 µm, 4.6 × 150 mm) from Tosoh Bioscience (Tokyo, Japan) on a Vanquish UHPLC system, controlled by Chromeleon version 7.2 (both Thermo Fisher Scientific, Waltham, MA, USA). Chromatography was carried out at 50 °C with mobile phase A (mpA) containing 0.5% TFA in water and mobile phase B (mpB) containing 0.4% TFA in ACN. Samples were injected without prior adjustment to the mobile phase composition during injection and adsorption. The chromatographic separation was carried out with a 10 min adsorption step applying 92% mpA and 8% mpB, followed by an 18 min linear AB gradient elution to 100% mpB and a 7 min re-equilibration with 92% mpA and 8% mpB at a flow rate of 0.5 mL/min. Eluting components were detected with a diode array detector, evaluating peak areas at 260 nm and 280 nm.

### 2.4. Analytics for Silica-SC-Based Extraction, RP-Based Extraction, and Quantification of Proteins and Nucleic Acids

For SDS-PAGE, NuPage 4–12% BisTris protein gels, LDS sample buffer, and MES running buffer were used and run on a PowerEase Touch 350 W Power Supply (all Invitrogen, Waltham, MA, USA) at reduced mode with 50 mM DTT in the sample solution. Protein staining was performed with a Coomassie blue solution. For NAGE, 0.7% agarose (Carl Roth, Karlsruhe, Germany) in TAE buffer (40 mM Tris, 20 mM acetic acid, 1 mM EDTA) with 1 µg/mL midori green (Nippon Genetics GmbH, Düren, Germany) was used and run on a PowerPac Basic (Bio-Rad, Hercules, CA, USA) followed by protein staining with a Coomassie blue solution of the gels, if necessary.

RiboGreen RNA and PicoGreen dsDNA assays (both Thermo Fisher Scientific, Waltham, MA, USA) were performed according to the manufacturer’s manual with minor adaptations. The assays were standardized with nucleic acids included in the respective assay kit and three replicates of six concentrations (0, 10, 30, 100, 300, 1000 ng/mL), if not stated otherwise. Samples were measured in three replicates of three dilutions (10-, 50-, and 100-fold). In the study outlined below, enzymatic treatment on a 50 µL scale was conducted prior to assay quantification to identify different nucleic acid species. Here, 2 U of DNase I (New England Biolabs, Ipswich, MA, USA) with reaction buffer provided by the DNase I manufacturer were used to degrade DNA for 10 min at 37 °C and heat inactivate for 10 min at 75 °C prior to the RiboGreen RNA assay. A mix of 2 µg RNase A and 5 U T1 (Thermo Fisher Scientific, Waltham, MA, USA) with reaction buffer (10 mM Tris-HCl, pH 7.5, 300 mM NaCl, 5 mM EDTA) was applied to degrade RNA for 30 min at 37 °C. With this RNase A/T1-Mix in corresponding reaction buffer in combination with 100 U S1 Nuclease (both Thermo Fisher Scientific, Waltham, MA, USA), both RNA and ssDNA were degraded prior to the PicoGreen DNA assay for 30 min at 37 °C. Nucleic acid concentrations determined by the RiboGreen assay were corrected according to the determined correlation between the observed and loaded nucleic acid masses [µg] (Figure 5a), according to Equation (1).
(1)$$Load\;NA_{hc}=\frac{Observed\;NA_{hc}}{0.9467}+0.2071\;µg$$

qPCR was conducted and analyzed with a QuantStudio 5 System (Thermo Fisher Scientific, Waltham, MA, USA). The PCR reaction comprised a 96 °C hold step for 7 min, followed by 40 cycles each at 96 °C for 15 s and 60 °C for 30 s. The primers used to amplify the dsDNA were 5′ TTCTTCAAGTCCGCCATGCCCG 3′ as the forward primer and 5′ TCGATGCCCTTCAGCTCGATGC 3′ as the reverse primer. Reactions were conducted with a final volume of 20 µL containing 2 µmol of each DNA polymerase primer, SYBR^®^Green (Bio-Rad Laboratories, Hercules, CA USA), and a 5 µL sample in a MicroAmp™ Optical 96-Well Reaction Plate (Thermo Fisher Scientific, Waltham, MA, USA). The analysis was standardized with dsDNA of known concentration, determined by PicoGreen assay, and three replicates of six dilutions (1-, 10-, 100-, 1000-, 10,000-, 100,000-fold). Samples were measured in three replicates of three dilutions (100-, 1000-, and 10,000-fold). Nucleic acid concentrations determined by qPCR can be corrected according to the determined correlation between observed and loaded dsDNA masses [µg] (Figure 5b), according to Equation (2).
(2)$$Load\;NA_{ther}=\frac{Observed\;NA_{ther}}{0.9592}+0.3452\;µg$$

To determine VLP morphology at RP-HPLC injection and for transmission electron microscopy (TEM) analysis, CC-purified VLPs were diluted with purification buffer and filtered through a 0.2 µm syringe filter. The VLPs were mixed 50:50 with either purification buffer for the control sample or solutions containing ACN and TFA to obtain ACN concentrations of 4%, 6%, and 8%, where 8% corresponds to the conditions of the adsorption step with 92% mpA and 8% mpB. VLP concentrations were set to 0.75 g/L for every condition. Samples were analyzed by TEM on a Fecnei Titan^3^ 80–300 microscope (FEI company, Hilsboro, OR, USA). Sample preparation was conducted according to studies with chimeric HBcAg VLPs [21] with 1% (*w*/*v*) alcian blue 8GX (Alfa Aesar, Ward Hill, MA, USA) in 1% acetic acid as hydrophilization solution and 2% ammonium molybdate (VI) (Acros Organics, Geel, Belgium) solution (at pH 6.25) as staining solution.

Protein and nucleic acid concentrations analyzed by RP-HPLC and UV/Vis were calculated from the peak area obtained and a conversion factor determined by calibration with Cp149 and dsDNA of known concentrations.

## 3. Results

### 3.1. Silica-SC-Based Extraction

For an accurate quantification of nucleic acids bound to HBcAg VLP proteins by dye-based fluorescence assays, a high recovery of the nucleic acids during silica spin column (SC)-based extraction and high purity of the obtained extracts are needed. Consequently, a silica-SC-based extraction procedure including proteinase K lysis followed by silica-SC chromatography was investigated for its use as a pre-treatment for the absolute quantification of NA_hc_ and NA_ther_ bound to HBcAg VLPs by dye-based fluorescence assays.

#### 3.1.1. Lysis of VLPs by Proteinase K

Enzymatic lysis times of 15 min and 60 min were investigated to achieve a high purity of nucleic acids after extraction and, hence, to enable accurate quantification of the nucleic acids present. For this, the silica-SC-based extraction was carried out with Cp164 after re-dissolution and an approximate total protein concentration of 4 g/L, determined by microvolume UV/Vis absorbance measurements. With a lysis time of 15 min, protein impurities could be detected in the samples after silica-SC-based extraction on SDS-PAGE (see lane 3, Figure 2). For a lysis time of 60 min, a considerably lower intensity of the impurity bands on the SDS-PAGE gel in the extracts after silica-SC-based extraction was observed. No protein contaminants could be determined by SDS-PAGE when VLP samples were diluted to approximately 25 ng/µL nucleic acids prior to the extraction (Figure Figure 3b). As a result, further silica-SC-based extraction experiments in this study were conducted with a proteinase K lysis time of 60 min.

#### 3.1.2. Adsorption of Nucleic Acids

In order to achieve efficient adsorption of the nucleic acids into the silica-SC matrix. Varying flow-through recirculation procedures were investigated. The silica-SC-based extraction was carried out with Cp164 after re-dissolution and 20-fold diluted to comply with the binding capacity of 5 µg for the silica-SC used. The elution of nucleic acids bound to the silica-SC was achieved using 100 µL of water. The extraction results from NAGE analysis are depicted in Figure 3a. Samples after silica-SC-based extraction without recirculation of the flow-through are shown in lanes 3 and 4 (samples were analyzed in duplicates), 5-fold recirculation in lanes 5 and 6, and 20-fold recirculation in lanes 7 and 8. In summary, an increase in the detected amount of nucleic acids in the extract after silica-SC-based extraction is observed when using recirculation procedures. The 5-fold recirculation procedure resulted in the highest fluorescence associated with nucleic acids (and presumably the amount of nucleic acids) on the NAGE. The corresponding final flow-through samples were analyzed on NAGE to support mass balance by displaying nucleic acids not bound to the silica-SC used in the flow-through fractions. Nucleic acids were, however, not detected in the flow-through fractions for all adsorption procedures. All investigated flow-through fractions produced high fluorescence signals in the gel pockets. However, nucleic acids that may be trapped in the gel pockets may be untraceable due to interfering fluorescence signals from other components in the flow-through fractions. In particular, analysis of the components present in the flow-through fractions revealed stained areas around the gel pockets for proteinase K and the binding buffer used (see Appendix A). Moreover, the extracts after silica-SC-based extraction were examined for protein contamination by SDS-PAGE. No protein contaminants were detected in the extracts; see lanes 3, 4, and 5 (Figure 3b, duplicates in Appendix A).

#### 3.1.3. Elution of Nucleic Acids

For an absolute quantification of the total amount of nucleic acids, an efficient elution of the nucleic acids bound to the silica-SC matrix is crucial. The silica-SC-based extraction was conducted with a load of 0.815 µg of nucleic acids (as listed in Appendix A), and the bound nucleic acids were subjected to varying elution conditions comprising 4 × 25 µL, 8 × 25 µL, and 1 × 200 µL elution volumes. Even though the procedure of applying a 1 × 200 µL elution volume resulted in a lower intensity of the nucleic acid band on the NAGE gel (lane 4 in Figure 4), compared to the initial load material in lane 1, it showed the highest intensities on NAGE as well as the highest nucleic acid recovery of 77.78 ± 0.93% when comparing the different elution modes. The 4 × 25 µL and 8 × 25 µL elution modes produced lower intensities on the gel, correlating with the lower recoveries measured by RiboGreen (Figure 4). The PicoGreen analysis was conducted in parallel but showed the same trends. Hence, the best quantitative silica-SC-based extraction in terms of purity and recovery was obtained with a 60 min lysis time, a 5-fold recirculation of the flow-through, and a 1 × 200 µL elution.

### 3.2. Characterization of the Silica-SC-Based Extraction

#### 3.2.1. Nucleic Acid Recovery

To investigate the influence of the amount of nucleic acids loaded onto the silica-SC during silica-SC-based extraction, the loadings of host cell-derived nucleic acids (NA_hc_) and dsDNA were varied. The observed masses in the extracts analyzed by RiboGreen for the varying amounts of processed NA_hc_ and the respective final recoveries are depicted in Figure 5a. The PicoGreen analysis was conducted in parallel but showed the same trends. With a higher amount of loaded NA_hc_ an increased proportion of nucleic acids is lost during silica-SC-based extraction. This is reflected by the gap between the identity line and the observed eluted masses. However, the recoveries rise from 55% for a 0.47 µg load to 81% for a 1.4 µg load to 85% for a 2.8 µg load, and up to 93% for a 4.67 µg load. The results for dsDNA are displayed in Figure 5b. The dsDNA behavior is similar to that of the NA_hc_, with loss of nucleic acids increasing with higher amounts of loaded dsDNA, except for a load of 0.95 µg dsDNA. Recovery rates range from 64% for a 0.31 µg load up to 86% for a 3.12 µg load of dsDNA. For a load of 0.95 µg dsDNA, however, a loss of 0.74 µg was observed, resulting in a recovery of 21%, substantiating that this measurement was an outlier. An overview of recovery rates observed in preliminary experiments and data from this work for various loadings of NA_hc_ and dsDNA is provided in Appendix A. Using the recoveries obtained for different loadings of nucleic acids during the silica-SC-based extraction, the observed nucleic acid concentrations can be corrected, as explained in Section 2.4.

#### 3.2.2. Quantification with and without Prior Silica Spin Column-Based Extraction

To demonstrate the need for prior nucleic acid extraction, nucleic acid quantification of Cp157 was performed by means of the RiboGreen assay, both with and without prior silica-SC-based extraction. Extraction was performed in biological triplicates for Cp157, and samples were analyzed by RiboGreen in four dilutions (undiluted, 10-, 50-, and 100-fold). The nucleic acid concentrations and relative standard deviations determined are listed in Table 2. The concentration without prior silica-SC-based extraction was determined to be 13.83 ± 5.97 ng/µL. In the case of prior extraction of nucleic acids, however, RiboGreen analysis yielded significantly higher concentration values of around 26.64 ng/µL for Cp157, with small differences between the biological replicates. The relative standard deviation of measurements without prior silica-SC-based extraction was high and equalled 43.17%, whereas the standard deviations in the case of prior extraction ranged between 3.58% and 6.75%. Similar results for preliminary silica-SC-based extraction procedures can be found in Appendix A.

### 3.3. Analysis of Extracted Nucleic Acids

To investigate differences in the fluorescent emissions of different nucleic acid types, lambda DNA (included in the PicoGreen assay) and 16S RNA (included in the RiboGreen assay) were analyzed with RiboGreen without any prior enzymatic treatment. Figure 6 presents the fluorescent emissions of lambda DNA and 16s RNA for 1, 10, 100, and 1000 ng/mL. Lambda DNA induces a higher fluorescent emission than 16s RNA with RiboGreen dye. The calculated regression lines show a slope of 0.93 AU*µL/ng for lambda DNA and a smaller slope of 0.24 AU*µL/ng for 16S RNA in this experiment.

To evaluate the influence of an enzymatic treatment on measurement accuracy and the composition of NA_hc_ in Cp164 VLPs, several enzymatic treatments were conducted after the silica-SC-based extraction but prior to PicoGreen and RiboGreen assay analysis. DNase I was used to degrade ssDNA and dsDNA prior to a RiboGreen RNA assay, while a RNase A/T1-Mix was used to degrade RNA, and a RNase A/T1-Mix in combination with a S1 Nuclease was used to degrade both RNA and ssDNA prior to a PicoGreen DNA assay. RiboGreen measurement of the extracted nucleic acids without an enzymatic treatment resulted in a total nucleic acid concentration of 220.20 ± 4.55 ng/mL, whereas 121.52 ± 2.21 ng/mL was determined by PicoGreen. Enzymatic treatments prior to analysis by dye-based fluorescence assays resulted in 64.4% RNA, 56.7% ssDNA/dsDNA, and 81.4% dsDNA lower nucleic acid concentrations compared to the respective analyses without enzymatic treatment (Table 3). For the quantification of specific nucleic acid species, a suitable quantification assay and enzymatic treatment are required after the silica-SC-based extraction.

**Figure 6 viruses-16-00013-f006:**
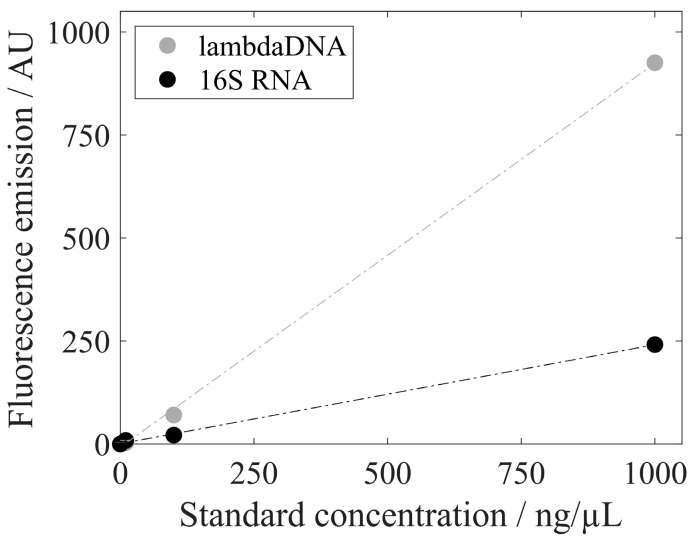
Fluorescence emission with RiboGreen dye of lambda DNA (standard solution provided by the PicoGreen assay kit) and 16S RNA (standard solution provided by the RiboGreen assay kit) with standard concentrations of 1, 10, 100, and 1000 ng/µL. Standard deviations are included but not visible.

### 3.4. Quantification of Proteins and Nucleic Acids Using RP-HPLC

#### 3.4.1. Lysis of VLPs

In contrast to the silica-SC-based extraction described above, the newly established RP-based extraction does not require a separate lysis step using proteinase K. Although samples were injected without prior adjustment to the mobile phase composition, the buffer components necessary for the RP-HPLC, namely ACN and TFA, were able to replace the proteinase K step. This was investigated using transmission electron microscopy (TEM). TEM analysis of VLPs treated with ACN and TFA to mimic RP-HPLC injection conditions revealed intact VLPs without ACN and TFA in the solution and decomposed VLPs in the presence of ACN and TFA (Figure 7).

#### 3.4.2. Development of RP-HPLC for Extraction and Quantification of VLP Proteins and Nucleic Acids

RP-HPLC was investigated as an integrated technique combining the separation of proteins and bound nucleic acids with concomitant quantification. Several HPLC columns with different combinations of mobile phases were screened in preliminary experiments (Appendix A), and the most suitable column was evaluated in terms of linearity, precision, and accuracy. The RP-HPLC chromatogram in Figure 8 displays the separation of nucleic acids from the sample in the flow-through (average A260/A280 of 1.78) and the gradient elution of the proteins from the sample (average A260/A280 of 0.59) with increasing the ACN concentration of the mobile phase. Parallel to the RP method, HPLC control runs without a prefilter and column were performed with Cp157 at three different dilution levels. Total peak areas at A280 of the control runs and RP method runs were compared, and recovery rates between 96.8% and 100.6% were found. No pressure rise was observed in RP-HPLC throughout the injections for all HBcAg constructs and purity levels.

#### 3.4.3. Method of Performance Assessment

The linearity of the RP protein quantification method was determined with Cp149 standards of known concentrations and resulted in a linear response between 0.29 and 9.27 µg with an R^2^ of 0.99. Within that range, Cp157 samples resulted in linear responses with peak areas for both the gradient (protein) and flow-through (nucleic acid). The precision of the method was determined by injecting Cp149 and Cp157 samples in triplicate on three different days (Table 4). The relative standard deviation (RSD) of the total peak area within each day varied between 1.93% (Cp149, Day 1) and 0.18% (Cp157, Day 2). The RSD of the peak area averaged over three days (intermediate precision) was 3.9% for Cp149 and 1.26% for Cp157. The accuracy of the RP method was determined by analyzing the same sample at different dilution stages. For Cp149, four different dilutions in the linear range were analyzed. The accuracy, the percentage of measured mass compared to the theoretical mass, ranged between 99% and 106% (Table 5).

VLP samples with varying purity levels were analyzed with the RP-HPLC method. Retention of HCPs in the RP column differed from VLPs within the elution step, enabling a purity determination of VLP protein to total protein. This analysis resulted in protein purities ranging widely from 17% to 65% for re-dissolved VLPs, while further purified VLP samples after the CaptoCore 400 (CC) purification step showed higher purities from 76% to 96%. Six different CC-purified VLP constructs with variable lengths of the nucleic acid binding region and with varying amounts of bound host cell-derived nucleic acids were analyzed with respect to nucleic acid and protein concentration. This resulted in nucleic acid to protein mass ratios of 0.0066 for Cp149, increasing values for constructs with longer nucleic acid binding regions, and leveling values around 0.2080 for Cp164, Cp167, and Cp183 (Figure 9). To compare nucleic acid quantification by silica-SC-based extraction with subsequent RiboGreen analysis and RP-based extraction with subsequent UV/Vis analysis, the six different CC-purified VLP constructs were analyzed with both methods. For all constructs, the nucleic acid concentration in the samples determined by RP-HPLC and UV/Vis was higher than the value obtained by silica-SC-based extraction and RiboGreen (Figure 10). When correcting the nucleic acid concentrations obtained for the silica-SC-based extraction and RiboGreen assay according to Equation (1), the resulting nucleic acid concentrations for Cp154, Cp157, and Cp167 samples are within the error of the fluorescent assay with maximum deviations of 11.5%. Whereas the results are outside the error of the fluorescent assay for Cp149, Cp164, and Cp183. However, Cp164 seems to be an outlier for both the silica-SC-based extraction with subsequent RiboGreen analysis and the RP-based extraction with subsequent UV/Vis analysis.

## 4. Discussion

### 4.1. Absolute Quantification of HBcAg Protein and Bound Nucleic Acids

The naturally occurring nucleic acid binding region of HBcAg is often used for effective therapeutic nucleic acid (NA_ther_) encapsulation [8,10]. During intracellular formation of the VLPs in *E. coli* [14], host cell-derived nucleic acids (NA_hc_) are encapsulated. To evaluate the removal of the bound nucleic acids and possible HBcAg protein loss during removal as well as the effective loading of VLPs with NA_ther_ later in the downstream process, precise absolute quantification of HBcAg VLPs and bound nucleic acids is essential.

In this work, a silica spin column (SC)-based extraction procedure, including enzymatic lysis and silica-SC chromatography, was investigated to attain high purity and recovery of the nucleic acids to be quantified, which is required for accurate quantification. The high purity of the nucleic acid extracts after silica-SC-based extraction was achieved by an extended proteinase K lysis time. The sufficient lysis of the VLP proteins ensures adequate separation of VLP proteins and bound nucleic acids. The recovery of nucleic acids was increased by recirculating the flow-through during the binding step and by increasing the elution volume in order to obtain the most accurate quantification of nucleic acids. Whereas in the literature, the purity of the extracts after silica-SC-based extraction and the recovery of the nucleic acids to be quantified were not evaluated for the extraction procedure reported for RNA in murine leukemia VLPs [30]. The investigation of varying loadings of nucleic acids in the silica-SC in our work revealed a relative correlation between the load and recovery of the nucleic acids, which was independent of the nucleic acid species (unspecific NA_hc_ or dsDNA, as model NA_ther_). Despite the higher absolute loss of nucleic acids during the silica-SC-based extraction, the relative recoveries increase with higher amounts of loaded nucleic acids. This means that high loads of nucleic acids are advantageous for the most precise quantification of bound nucleic acids. However, care needs to be taken not to exceed the binding capacity of the SC used. As this can be challenging due to imprecise approximations of protein and nucleic acid concentrations by means of existing techniques, the assessment of the recoveries for a range of nucleic acid loads enables a correction of the nucleic acid concentrations observed by silica-SC-based extraction and dye-based fluorescence measurements to the actual amounts of loaded nucleic acids in the sample, as implemented here.

Analysis of HBcAg VLPs with bound NA_hc_ with and without prior nucleic acid extraction clearly demonstrated the value of the silica-SC-based extraction. Nucleic acid quantification by RiboGreen provided consistent results with negligible standard deviations between the biological replicates (*n* = 3). Analysis of the same sample without prior silica-SC-based extraction yielded smaller nucleic acid concentration values compared to the results with prior silica-SC-based extraction. The standard deviation of nucleic acid concentrations significantly exceeded those determined with a prior extraction. This might also explain the conflicting results in the literature between UV-spectroscopy-based analysis (A260/A280) and a dye-based fluorescence assay without a prior separation of the HBcAg VLPs and bound NA_hc_ [27]. However, the analysis of DNA impurities by the PicoGreen assay in HBcAg VLP purification [14] does not seem to be affected by a lack of nucleic acid extraction, as the DNA presumably was not bound to the VLPs and binding of the dye to the DNA was not sterically impeded. This emphasizes the need for effective nucleic acid extraction to accurately quantify nucleic acids bound to HBcAg VLPs.

We further analyzed the NA_hc_ obtained by the silica-SC-based extraction procedure to evaluate the need for complex enzymatic treatments prior to analysis and the use of different specific fluorescent dyes to analyze different nucleic acid species. As expected, the lambda DNA produced a fluorescence signal other than that of the 16S RNA included in the RiboGreen assay. This clearly demonstrates the need for an enzymatic degradation step prior to the precise quantification of a specific nucleic acid species by dye-based fluorescence assays, as outlined by the assay manufacturer. In our study, the lambda DNA fluorescence signal of the RiboGreen dye was much higher than that of the 16S RNA provided in the assay kit, consistent with the stronger fluorescence enhancement of RiboGreen dye binding to dsDNA than for binding to an equal mass of RNA described in the literature [38]. Further, this correlates with the results for the host cell-derived nucleic acid composition, assuming different fluorescence signals for dsDNA and ssDNA. Results for RiboGreen and PicoGreen assays differ significantly for samples with and without enzymatic treatment, indicating a complex interplay of fluorescence signals from the different nucleic acid species in the untreated samples. To quantify specific nucleic acid species, a suitable enzymatic pre-treatment for dye-based fluorescence assay quantification seems inevitable. This might also be useful when evaluating the removal of specific NA_hc_ from HBcAg VLPs, as in RNA-specific lithium chloride precipitation [8] or enzymatic treatments [18,39]. For the evaluation of general nucleic acid depletion [15,16,27]), it may be sufficient and advantageous in terms of time and costs to perform the silica-SC-based extraction procedure in combination with a dye-based fluorescence assay such as PicoGreen or RiboGreen, but without an enzymatic treatment. Depending on the objective of the specific experiment and the expected nucleic acid species, the complexity of the sample treatment between the silica-SC-based extraction proposed here and the absolute quantification of nucleic acids by dye-based fluorescence assays can be varied and adapted to the specific purpose.

The RP-based extraction developed separates HBcAg proteins from bound nucleic acids and enables concomitant absolute quantification of both proteins and nucleic acids in the samples by UV/Vis. Unlike other RP-HPLC methods for the quantification of VLP proteins [14,24,25], no dissociating pre-treatment of the samples is required before injection. The often-used guanidine-HCl dissociation reagent [14,24] is visible in the UV spectra and overlaps with the nucleic acid peaks in the flow-through of the chromatograms, which impedes nucleic acid quantification. However, injection of the samples into the RP-HPLC buffer containing ACN and TFA, even without prior adjustment of the samples to the mobile phase composition, appears to be sufficient to effectively dissociate the VLPs. This prevents blockage of the chromatography column and additionally separates the bound nucleic acids from the HBcAg proteins. In contrast to other HPLC-based quantification methods for HBcAg VLPs [19,20,21], the HBcAg proteins and bound nucleic acids are separated and can therefore be accurately quantified by their UV spectra. Having evaluated the critical performance parameters with results comparable to those presented in the literature [25], it can be stated that the RP-HPLC method developed can be used to reliably quantify the absolute amounts of HBcAg proteins and bound NA_hc_. The method has not yet been tested with NA_ther_, but it is expected to be applicable to nucleic acids of a certain length. Investigations of different levels of purity of HBcAg VLPs with bound nucleic acids during the downstream purification process demonstrated the method’s potential to determine HBcAg VLP purities. Due to varying retention times for HCPs and HBcAg protein, the method can further be applied to analyze intermediate purification samples and to support purification process development.

The RP-based extraction was applied to six different HBcAg constructs with different lengths of the naturally occurring nucleic acid binding region. The mass ratios of the measured absolute amounts of nucleic acids and proteins correlate with the loading results of the HBcAg constructs obtained from the relative analysis of the A260/A280 ratio described in an earlier publication by our group [26]. Depending on the length of the nucleic acid binding region and the positive charges present, different amounts of nucleic acids are encapsulated in the HBcAg VLPs [26,40]. In Cp183, with around 3600 positive charges in the arginine-rich binding region of one HBcAg VLP capsid, 4970 nucleotides in the form of RNA per capsid were determined earlier by a simplified UV-spectroscopy method [28]. This results in a nucleic acid-to-protein mass ratio of around 0.34 when calculating with a molecular weight of 4.8 MDa for a Cp183 HBcAg capsid (*T* = 4) and 326.99 g/mol for a nucleotide. However, the lower mass ratio of 0.22 for Cp183 obtained by RP-HPLC in our study suggests a lower load of HBcAg VLPs with nucleic acids compared to the literature [28]. A comparison of the results for nucleic acid quantification by silica-SC-based extraction and subsequent RiboGreen quantification and the RP-based extraction and quantification revealed discrepancies in the absolute nucleic acid concentration in the samples for the six different HBcAg constructs. After a correction procedure using Equation (1), however, the concentrations calculated by the RP-HPLC method are mainly within the error of the corrected nucleic acid concentrations obtained by RiboGreen. The residual differences between the concentrations may be explained by operating errors and could be prevented by performing silica-SC-based extractions and RP-HPLC injections in replicates. The dye-based fluorescence nucleic acid quantification assays are highly sensitive, with a minimum detectable mass of, for example, 200 pg of RNA for the RiboGreen assay, as stated by the manufacturer. The sensitivity of the UV/Vis analysis to nucleic acids is dependent on the path length of the sample cell and instrument settings, such as, e.g., scan speed. We assume a sensitivity in the low ng range for the analytical setup used in this study. However, the comparison of the two methods presented for nucleic acid quantification showed that sensitivity was not a concern, with values in the nanogram nucleic acid range. Both methods are applicable to quantify nucleic acids bound to HBcAg VLPs in the concentration ranges present during process development. For a more specific and sensitive analysis, if needed, the nucleic acids in the flow-through of the RP-HPLC methods could be fractionated and analyzed further by the described dye-based fluorescence assay analyses.

### 4.2. Analytical Toolbox for HBcAg VLPs

The here-investigated silica-SC-based extraction, followed by adaptable quantification with and without prior enzymatic treatments, is a powerful tool to precisely quantify different species of nucleic acids bound to HBcAg VLPs by dye-based fluorescence assays. However, protein lysis prevents protein quantification in the flow-through fractions of the silica-SC chromatography procedure by, e.g., UV/Vis-based methods due to protein degradation. The RP-based extraction developed enables simultaneous absolute quantification of both HBcAg proteins and bound nucleic acids by UV/Vis spectroscopy. However, detailed nucleic acid species determination is lacking. Both methodologies overcome the limitations of existing techniques, and effective absolute quantification of HBcAg proteins and bound nucleic acids is added to the analytical toolbox for HBcAg VLP research. Today, only qualitative analytical techniques such as SDS-PAGE for proteins and NAGE for nucleic acids and proteins, in combination with microvolume UV/Vis spectroscopy measurements for approximations of protein and nucleic acid concentrations, are used for HBcAg VLP process development for gene delivery [8,10,15,16,27]. Size exclusion HPLC is also frequently used to separate and quantify different HBcAg VLP protein species (capsids and dimers) and impurities such as proteins and free nucleic acids [19,20,21]. However, it cannot separate nucleic acids bound to the HBcAg VLPs. The combination of these analytical tools with the two methods presented for absolute quantification of HBcAg VLPs and bound nucleic acids may be used to more accurately evaluate (i) the removal of bound NA_hc_ [8,10,15,16,18,27], (ii) possible HBcAg protein loss during the removal, and (iii) the effective loading of the VLPs with NA_ther_ [10,16] based on, e.g., encapsulation efficiency or payload of the VLPs.

## 5. Conclusions

We investigated a nucleic acid extraction procedure for the accurate absolute quantification of NA_hc_ or NA_ther_ bound to HBcAg VLPs by dye-based fluorescence assays. The silica-SC-based extraction revealed load-dependent nucleic acid recoveries, which allowed for a back calculation of initial nucleic acid concentrations. Moreover, we developed an RP-based extraction to separate and simultaneously quantify the HBcAg VLP proteins and bound nucleic acids by RP-HPLC. The method was successfully evaluated using the performance parameters of linearity, precision, and accuracy. It was applied to six different HBcAg VLP constructs with different amounts of encapsulated nucleic acids and found to be suited for determining absolute protein and nucleic acid concentrations as well as HBcAg protein purities. Both methods overcome the limitations of existing analytical techniques and, in combination with already existing analytical tools, can support gene delivery process development for HBcAg VLPs.

## Figures and Tables

**Figure 2 viruses-16-00013-f002:**
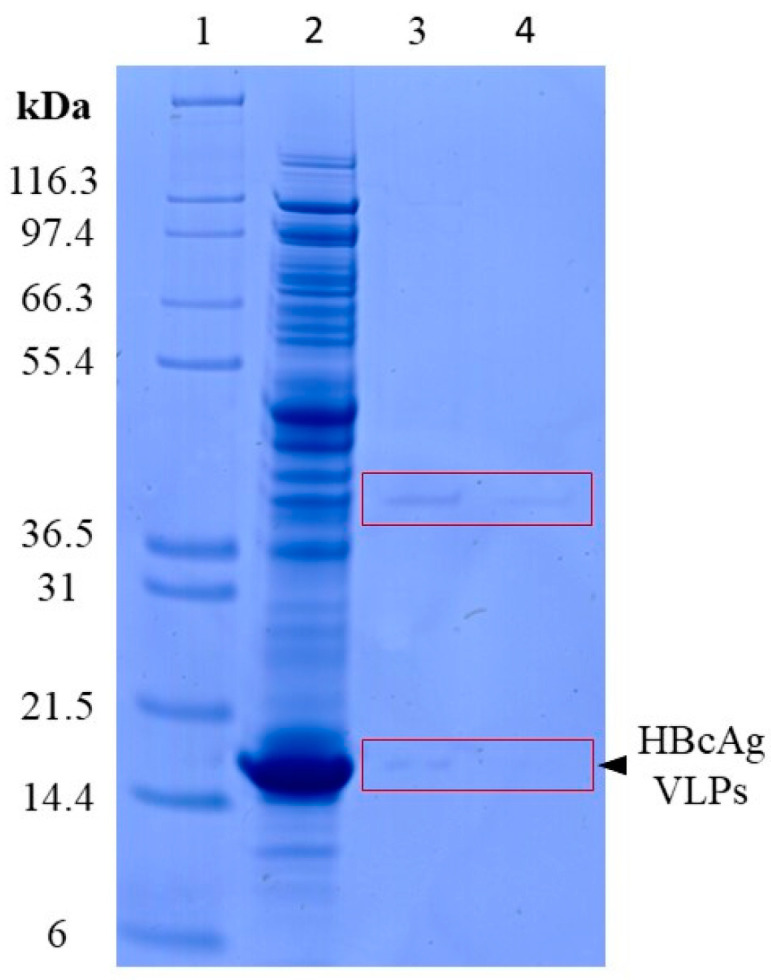
Investigation of proteinase K lysis times. Remaining protein impurities after silica-SC-based extraction for 15 min (lane 3) and 60 min (lane 4) lysis time on SDS-PAGE for undiluted Cp164. Marker in lane 1 and the initial Cp164 VLP sample before extraction in lane 2. The boxes were added to guide the eye. HBcAg: Hepatitis B core Antigen; SC: spin column; VLP: virus-like particle.

**Figure 3 viruses-16-00013-f003:**
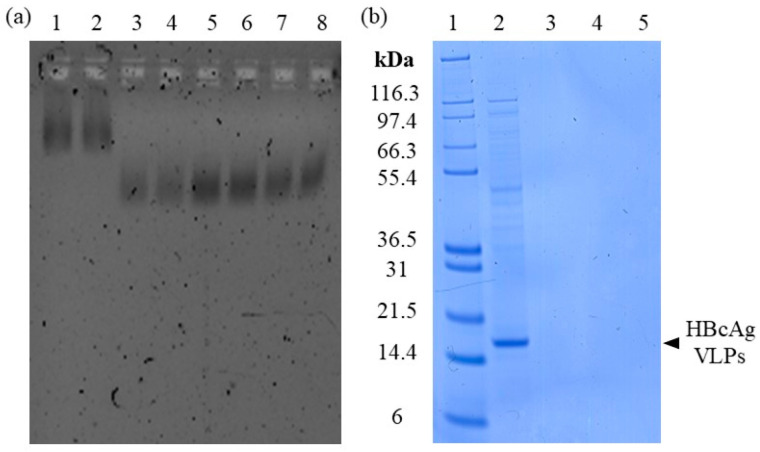
Investigation of silica-SC adsorption. (**a**) Nucleic acids on NAGE in the extracts after silica-SC-based extraction. Depicted are extracts without recirculation of the flow-through in lanes 3 and 4, extracts with recirculation of the flow-through fivefold (lanes 5 and 6) and 20-fold (lanes 7 and 8), and nucleic acids bound in the initial Cp164 VLP sample (diluted 20-fold) prior to extraction in lanes 1 and 2. (**b**) Detection of protein content in the extracts after silica-based extraction without recirculation and 5-fold and 20-fold recirculation in lanes 3, 4, and 5, respectively, on SDS-PAGE. Marker in lane 1 and initial Cp164 VLP sample (diluted 20-fold) before extraction in lane 2. NAGE: native gel electrophoresis; SC: spin column; VLP: virus-like particle.

**Figure 4 viruses-16-00013-f004:**
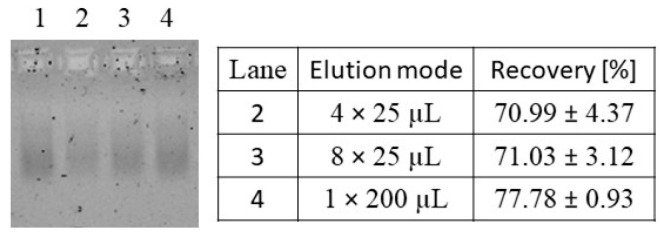
Investigation of the silica-SC elution procedure. Nucleic acids in the extracts after silica-SC-based extraction on NAGE and respective recovery values obtained by RiboGreen analysis for varying elution procedures were 4 times 25 µL (lane 2), 8 times 25 µL (lane 3), and 1 time 200 µL (lane 4). Initial load material of CC-purified Cp157 (diluted to comply with silica-SC binding capacity; see Section 2.2) in lane 1. CC: CaptoCore 400; NAGE: native gel electrophoresis; SC: spin column.

**Figure 5 viruses-16-00013-f005:**
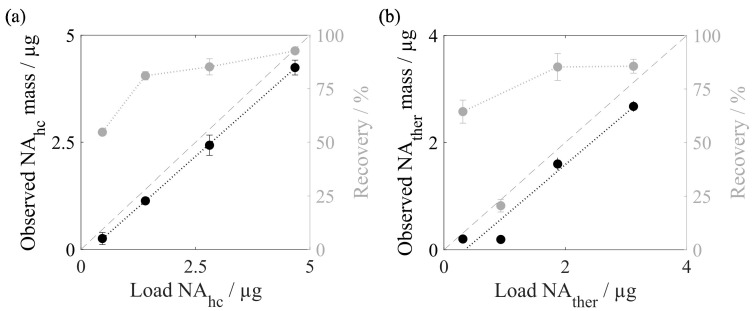
Nucleic acid recoveries for silica-SC-based extraction. (**a**) Observed NA_hc_ mass and identity line in black, and respective recoveries for different amounts of loaded NA_hc_ derived from Cp157 HBcAg VLPs in grey, analyzed by RiboGreen. (**b**) Observed dsDNA mass and identity line in black, and respective recoveries for different amounts of loaded dsDNA in grey, analyzed by qPCR. Lines were added to guide the eye. HBcAg: Hepatitis B core Antigen; NA_hc_: host cell-derived nucleic acids; NA_ther_: therapeutic nucleic acids; VLP: virus-like particle.

**Figure 7 viruses-16-00013-f007:**
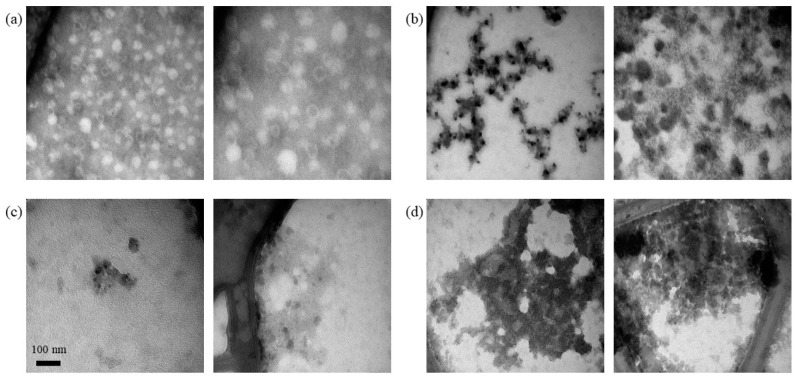
Lysis of VLPs by RP-HPLC injection conditions. TEM analysis of Cp157 HBcAg VLPs was diluted with purification buffer and mixed 50:50 with either purification buffer for the (**a**) untreated control sample or solutions containing ACN and TFA to obtain ACN concentrations of (**b**) 4%, (**c**) 6%, and (**d**) 8% and a VLP concentration of 0.75 g/L, respectively. ACN: acetonitrile; HBcAg: Hepatitis B core Antigen; TEM: transmission electron microscopy; TFA: trifluoroacetic acid; VLP: virus-like particle.

**Figure 8 viruses-16-00013-f008:**
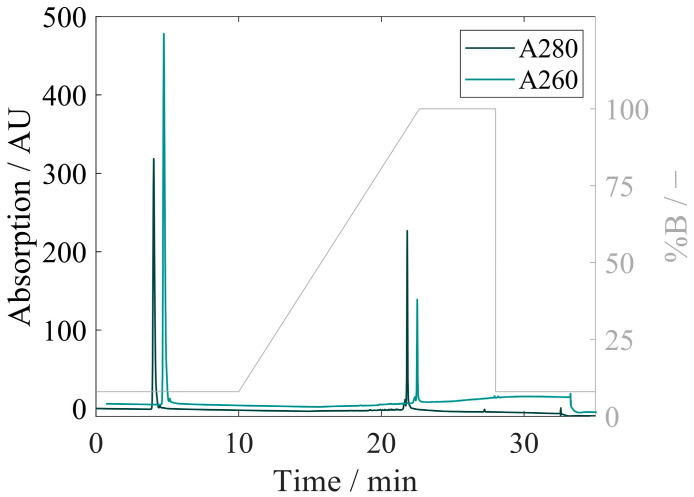
RP-HPLC chromatogram of the Cp157 HBcAg VLP sample with the A280 and A260 (for clarity, shifted +0.7 min in the x-direction and +75 AU in the y-direction) signals on the primary y-axis and the set percentage of buffer B in the mobile phase to highlight the RP-HPLC method phases on the secondary y-axis over the method run time. HBcAg: Hepatitis B core Antigen; HPLC: high-performance liquid chromatography; RP: reversed phase; VLP: virus-like particle.

**Figure 9 viruses-16-00013-f009:**
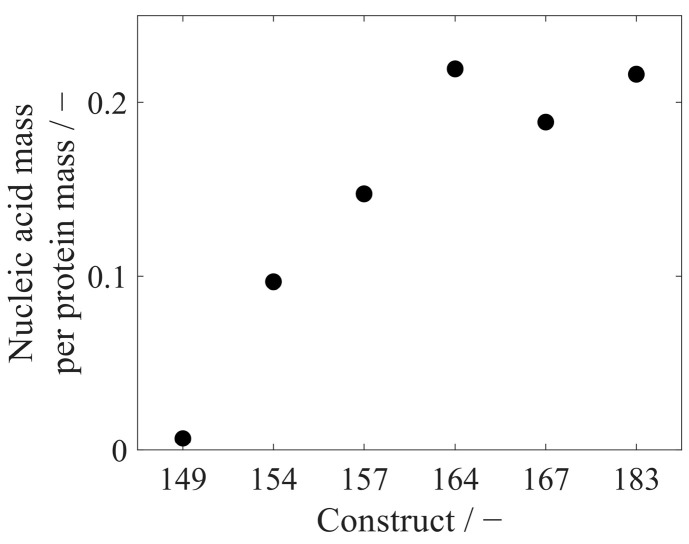
Mass ratio of nucleic acids to proteins for six different HBcAg VLP constructs. Absolute quantification was conducted by the developed RP-HPLC method in one replicate. HBcAg: Hepatitis B core Antigen; HPLC: high-performance liquid chromatography; RP: reversed phase; VLP: virus-like particle.

**Figure 10 viruses-16-00013-f010:**
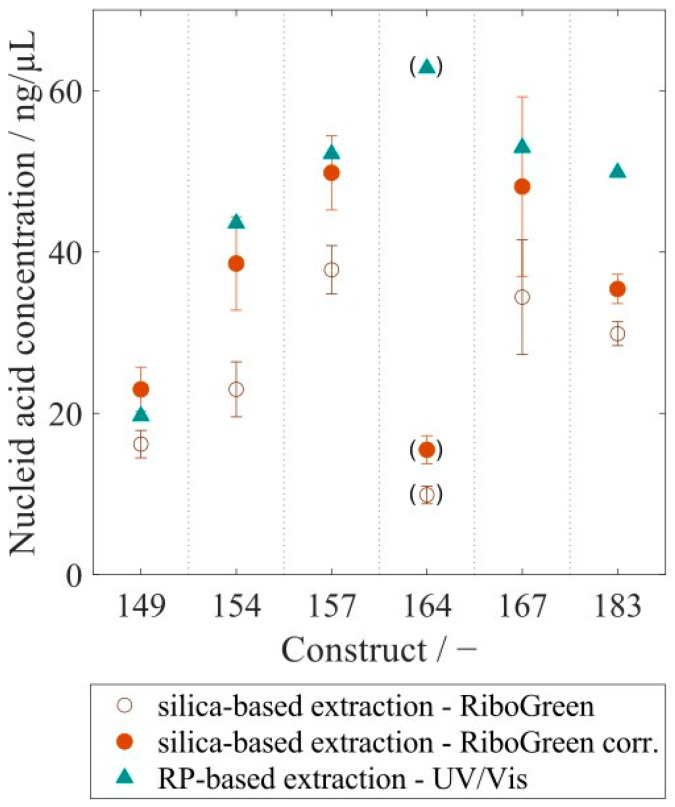
Comparison of nucleic acid quantification results obtained by RP-based extraction using UV-Vis and silica-SC-based extraction using a RiboGreen assay. The results of the RiboGreen assay were corrected, following the correction procedure using Equation (1) outlined in Section 4 and depicted in Figure 5a. RP: reversed phase.

**Table 2 viruses-16-00013-t002:** Results for absolute nucleic acid concentrations of Cp157 HBcAg VLPs samples without and with prior silica-SC-based extraction of nucleic acids (in triplicate), analyzed by the RiboGreen assay. HBcAg: Hepatitis B core Antigen; VLP: virus-like particle.

	Concentration in ng/µL	Relative Standard Deviation [-]
Cp157 without extraction	13.83 ± 5.97	43.17
Cp157 with extraction #1	27.02 ± 0.97	3.58
Cp157 with extraction #2	25.96 ± 1.75	6.75
Cp157 with extraction #3	26.94 ± 1.41	5.24

**Table 3 viruses-16-00013-t003:** Quantification of RNA and DNA species in Cp164 HBcAg VLPs after silica-SC-based extraction and treatment with different enzymes to identify different nucleic acid species using RiboGreen and PicoGreen. HBcAg: Hepatitis B core Antigen; VLP: virus-like particle.

Assay	Treatment	Determined Nucleic Acid Species	Concentration in ng/mL
RiboGreen	-	Total	220.20 ± 4.55
	DNAse 1 Treated	RNA	78.34 ± 1.67
PicoGreen	-	Total	121.52 ± 2.21
	RNAse A/T1	DNA	52.63 ± 3.60
	RNAse A/T1 + S1 Nuclease	dsDNA	22.57 ± 1.57

**Table 4 viruses-16-00013-t004:** RP-HPLC method precision assessed by total peak area at A280 for Cp149/Cp157. HPLC: high-performance liquid chromatography; RP: reversed phase; RSD: relative standard deviation.

Injection	Peak Areas at A280 Cp149/Cp157		
	Day 1	Day 2	Day 3
1	14.55/9.36	13.74/9.11	14.98/9.50
2	14.44/9.21	13.46/9.10	14.53/9.22
3	13.92/9.06	13.24/9.14	14.98/9.44
Average	14.31 ± 0.28/9.21 ± 0.12	13.48 ± 0.21/9.12 ± 0.02	14.82 ± 0.21/9.39 ± 0.12
Repeatability/RSD	1.93%/1.32%	1.53%/0.18%	1.42%/1.27%
Intermediate precision	3.9%/1.26%		

**Table 5 viruses-16-00013-t005:** RP-HPLC method accuracy assessed by total peak area at A280 at four concentrations of Cp149 HBcAg VLP. HBcAg: Hepatitis B core Antigen; HPLC: high-performance liquid chromatography; RP: reversed phase; VLP: virus-like particle.

Concentration Level	Measured Mass (µg)	Theoretical Mass (µg)	Accuracy
1	8.76	8.67	101%
2	4.48	4.54	99%
3	2.70	2.54	106%
4	1.68	1.59	106%

## Data Availability

The raw data supporting the conclusions of this article will be made available by the authors without undue reservation.

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
