# Peer review of "Absolute Quantification of Hepatitis B Core Antigen (HBcAg) Virus-like Particles and Bound Nucleic Acids"

_viruses, 2023, doi:10.3390/v16010013_

Round 1

Reviewer 1 Report

Comments and Suggestions for Authors

The authors of the manuscript "Absolute Quantification of HBcAg Virus-like Particles and Bound Nucleic Acids" describe the elaboration of new purification and quantification methods for nucleic acids encapsidated into virus-like particles.

As shown in the manuscript, encapsidated nucleic acids can be purified and quantified by reversed-phase high-performance liquid chromatography and by silica spin columns after proteinase K degradation of viral proteins.

Moreover, the authors demonstrate the application of elaborated methods by analyzing the RNA content in VLPs with variable lengths of the nucleic acid binding region and with varying amounts of bound host cell-derived nucleic acids. The experiments resulted in a clear correlation between the length of the nucleic acid binding region and the amounts of RNA in VLPs. The elaborated methods revealed that nucleic acids constitute 22% of the total HBcAg VLP mass.

Only some minor points could be identified in the manuscript:

Line 32: It is recommended to include a citation regarding the role of encapsidated host cell nucleic acids in immune stimulation and Toll-like receptor activation;

Line 169: Provide more detailed information about the protocol, including the amounts of proteinase K (in units) used for protein degradation;

Line 181: Clearly describe how the reference nucleic acid was obtained;

Line 217: Provide details on the conditions for DNAseI, RNAse A/T1 mix, and S1 nuclease;

Line 286: Specify the binding capacity of the silica column used;

Line 358: Add legends for the points/lines in Fig.5;

Line 421: Clarify whether the disassembly of VLPs in ACN/TFA buffer is complete or partial, or if VLPs simply aggregate in the presence of ACN/TFA.

In general, the study is acceptable and the results are sufficiently presented in the manuscript. The elaborated methods for nucleic acid quantification hold potential for technological developments based on virus-like particles. Therefore, with the correction of these minor points, the manuscript can be considered for publication.

Author Response

Dear Reviewer 1,

Thank you very much for reviewing the manuscript. We appreciate the positive feedback on our experimental work on VLP analytics. Any corrections and changes in the manuscript are marked in red.

Regarding your minor remarks:

Line 32: Thank you very much on your interest in the role of encapsidated host-cell derived nucleic acids in immune stimulation and Toll-like receptor activation. In the following publication authors observed a switch in the immune responses of Balb/C mice from a Th1 response, with the production of IgG2a antibodies, to a Th2 response, with the production of IgG1 antibodies, depending on the encapsidation ability due to the C-terminal truncation of the nucleic acid binding region: https://doi.org/10.1371/journal.pone.0075938 We included this article in the manuscript. However, the investigated HBcAg VLPs are prospected to be used for gene delivery and therefore the encapsulated host-cell derived are removed during the purification process and replaced by the therapeutic nucleic acids.

Line 169: Thank you very much for this hint. We included more detailed information about the protocol in the manuscript.

Line 181: Thank you for the comment on the production of the reference nucleic acid. The description about the dsDNA production can be found from line 163. We have tried to explain more clearly how the reference DNA was produced.

Line 217: Thank you for your comment on the conditions for the enzymatic treatment. We provided more details on the used conditions.

Line 286: Thank you very much for drawing attention to the unmentioned binding capacity of the silica column used. We included it in the description.

Line 358: Thank you for your suggestion to add legends for the points and lines in Figure 5. The added value of additional legends is not obvious to us. The assignment of the points is made clear by the use of two different colours for the data sets and the corresponding y-axis. An additional legend would, in our opinion, be redundant. However, we added a more detailed description on the data sets in the caption and hope this clarifies the allocation. Furthermore, the lines are explained in detail in the caption, if not obvious.

Line 421: Thank you very much on your interest in the VLP disassembly in ACN/TFA buffer. Whether the VLPs are partially or completely dissociated cannot be fully elucidated. Based on the transmission electron microscopy results and the fact that no pressure increase was observed in the RP-HPLC during the injections for all HBcAg constructs and purity levels, we assume that the VLPs are dissociated and not aggregated as this would cause pressure increase and eventually column blockage.

Reviewer 2 Report

Comments and Suggestions for Authors

Angela Valentic and colleagues present a high quality and well-written experimental manuscript focused on absolute quantification of HBcAg virus-like particles and bound nucleic acids. 

Authors investigated a silica spin column based extraction procedure, including proteinase K lysis and silica chromatography, for the absolute quantification of different species of nucleic acids bound to HBcAg VLPs analysed by dye-based fluorescence assays. This revealed load-dependent nucleic acid recoveries of the performed silica-SC based extraction. 

Authors also developed a reversed-phase high performance liquid chromatography method to separate and quantify the HBcAg proteins and the bound nucleic acids simultaneously without prior sample treatment by dissociation reagents. The method demonstrated sufficient linearity, accuracy, and precision coefficients, and is suited for determining absolute protein and nucleic acid concentrations and HBcAg protein purities at various purification stages. 

Authors found that both the silica-SC based extraction and RP based extraction presented overcome limitations of analytical techniques, which are restricted to relative or qualitative analyses for HBcAg VLPs with bound nucleic acids. In combination with existing analytics, the methods for an absolute quantification of HBcAg VLPs and bound nucleic acids presented here are required to evaluate downstream purification steps, such as the removal of host cell-derived nucleic acids, concurrent protein loss, and efficient loading with therapeutic nucleic acids. 

Finally, authors conclude that they investigated a nucleic acid extraction procedure for the accurate absolute quantification of NAhc or NAther bound to HBcAg VLPs by dye-based fluorescence assays. The silica-SC based extraction revealed load-dependent nucleic acid recoveries, which allowed for a back calculation to initial nucleic acid concentrations. They also developed an RP based extraction to separate and simultaneously quantify the HBcAg VLP proteins and bound nucleic acids by RP-HPLC. The method was successfully evaluated using the performance parameters of linearity, precision, and accuracy. Authors suggest that these methods are key for an effective process development when using HBcAg VLP as potential gene delivery vehicles.

Overall, the manuscript is highly valuable for the scientific community and should be accepted for publication.

======================

Other comments to authors:

1) Please check for typos throughout the manuscript.

2) Please improve figures/tables where appropriate

3) With regards to the nuclease activity – authors are encouraged to cite the following article that describes the capability analysis of endonuclease to hydrolyse DNA. DOI: 10.3389/fphar.2018.00114

Author Response

Dear Reviewer 2,

Thank you very much for reviewing the manuscript. We appreciate the positive feedback on our experimental work on VLP analytics. Any corrections and changes in the manuscript are marked in red.

Regarding your remarks:

  • Thank you very much for this hint. We carried out an additional round of spelling corrections.
  • Thank you very much for this comment. We re-evaluated the figures and tables for clarity and meaningfulness.
  • Thank you very much for drawing attention on the article that describes the capability analysis of endonuclease to hydrolyse DNA. We included it in the manuscript in line 568.

Best Regards.